# A Data-Challenge Case Study of Analyte Detection and Identification with Comprehensive Two-Dimensional Gas Chromatography with Mass Spectrometry (GC×GC-MS)

**Stephen E. Reichenbach** [1,*] **, Qingping Tao** [2]**, Chiara Cordero** [3] **and Carlo Bicchi** [3]

[1] Computer Science & Engineering Department, University of Nebraska, Lincoln, NE 68588-0115, USA
[2] GC Image, LLC, Lincoln, NE 68505-7403, USA
[3] Dipartimento di Scienza e Tecnologia del Farmaco, Università degli Studi di Torino, 10125 Turin, Italy
***** Correspondence: reich@unl.edu

**Abstract:** This case study describes data analysis of a chromatogram distributed for the 2019 GC×GC Data Challenge for the Tenth Multidimensional Chromatography Workshop (Liege, Belgium). The chromatogram resulted from chemical analysis of a terpene-standards sample by comprehensive two-dimensional chromatography with mass spectrometry (GC×GC-MS). First, several aspects of the data quality are assessed, including detector saturation and oscillation, and operations to prepare the data for analyte detection and identification are described, including phase roll for modulation-cycle alignment and baseline correction to account for the non-zero detector baseline. Then, the case study presents operations for analyte detection with filtering, a new method to flag false detections, interactive review to confirm detected peaks, and ion-peaks detection to reveal peaks that are obscured by noise or coelution. Finally, the case study describes analyte identification including mass-spectral library search with a new method for optimizing spectra extraction, retention-index calibration from preliminary identifications, and expression-based identification checks. Processing of the first 40 min of data detected 144 analytes, 21 of which have at least one percent response, plus an additional 20 trace and/or coeluted analytes.

**Keywords:** comprehensive two-dimensional gas chromatography (GC×GC); chemometrics; cheminformatics

## 1. Introduction

The difficulty of detecting and identifying analytes in data from comprehensive two-dimensional gas chromatography with mass spectrometry (GC×GC-MS) ranges from simple, for resolved analytes that have clear spectral signals matched in mass-spectral libraries, to challenging, for coeluted and trace analytes that have obscure or faint signals and ambiguous matching with mass-spectral libraries [1–5]. This case study examines a combination of time-tested and new peak detection techniques, beginning with the 2D drain algorithm that is highly effective for resolved peaks, filtering of those peaks, followed by a new method for predicting true and false peak detections, and combined with a new tool that detects collections of coincident ion-peaks. It then considers analyte identification with mass-spectral library search, using a new method for parameterizing extraction of mass spectra in order to maximize search performance; retention-index calibration, using library retention indices from preliminary identifications; and expression-based identification checks.

The data for the case study is a GC×GC-MS chromatogram of a terpene-standards sample released for the Data Challenge at the *10th Multidimensional Chromatography Workshop* (Liege, Belgium, January

2019) [6]. Using open data allows for broad examination of the data, analysis, and results. The data was acquired using a Leco Pegasus® 4D system (St. Joseph MI, USA) with a liquid nitrogen quad-jet modulator and time-of-flight (TOF) MS. The run time was 91.125 min with data acquisition delay of 8.75 min and a modulation cycle of 2.5 s. The system acquired 200 spectra per second with integer mass-precision and mass-to-charge (*m/z*) range 30–800. No other information about the chromatogram or its acquisition was provided. For the Data Challenge, the data was made available in both Leco's proprietary PEG format and the ASTM interchange-standard CDF format [7].

Figure 1 shows a two-dimensional (2D) image and pseudo-three-dimensional (3D) surface map of the total intensity count (TIC), produced with GC Image® GC×GC Software© V2.8 (Lincoln NE, USA) [8], which also was used in processing the data. No large peaks are present after about 42 min of first-dimension ($^1$D) retention time ($^1t_R$), although chromatographic artifacts could be detected until near the end of the run. Therefore, for more concise presentation and convenient visualization, only the first 40 min of data (from 8.75–48.75 min $^1t_R$), which contained all terpene standards and detected impurities, was imported for analysis and the imported *m/z* range was limited to 30–220, consistent with the analytes separated during that time, to improve the signal-to-noise ratio (SNR).

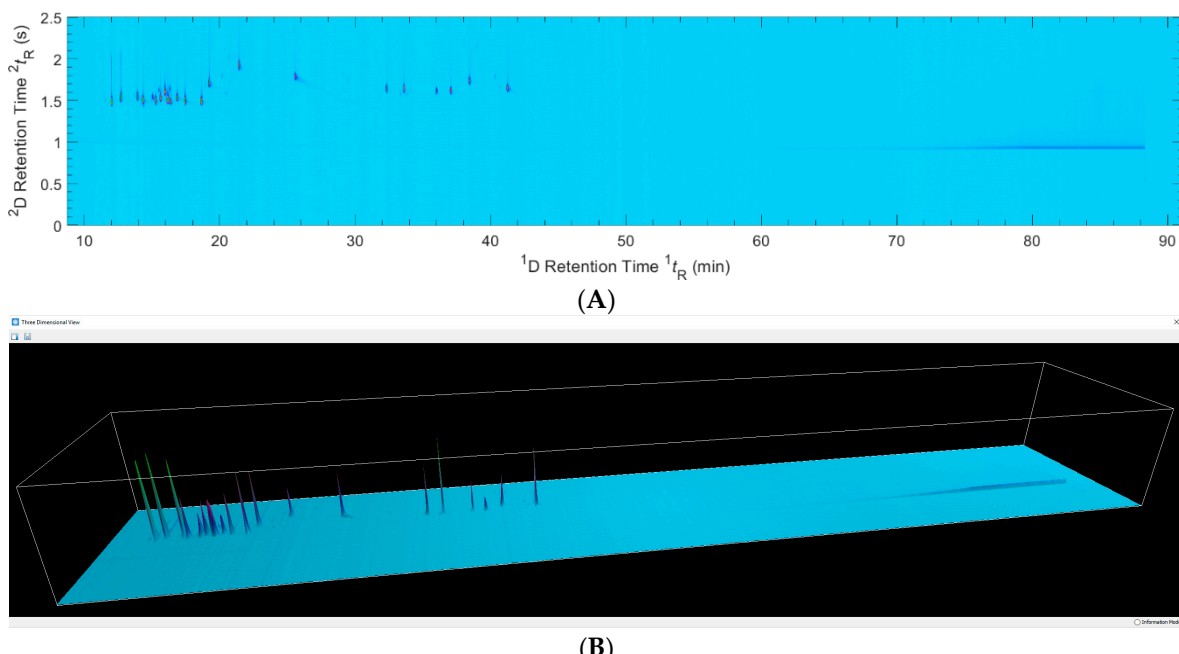

**(A)**

**(B)**

**Figure 1.** Terpene-standards GCxGC TIC chromatogram with pseudo-colorization using linear value mapping over the intensity range 78,895-3,560,720 DN, shown as: (**A**) A 2D image and (**B**) a pseudo-3D surface.

The case study is organized in three sections: Data quality and preprocessing, analyte detection, and analyte identification. First, several aspects of the data quality are assessed, including detector saturation and oscillation, and operations to prepare the data for visualization, peak detection, and identification are described, including phase roll for modulation-cycle realignment and baseline correction to account for the non-zero detector baseline. Then, blob detection with filtering delineates analyte peaks, a new method is used to flag potential false detections, detected peaks are confirmed by interactive spectral review, and ion-peaks detection is used to detect peaks that are obscured by noise or coelution. Finally, the analytes are identified putatively by mass-spectral library search using a new method for extracting mass spectra to optimize search performance, followed by first-column retention-index ($I^T$) calibration based on preliminary identifications and library retention-indices (*I*), then expression-based identification checks.

## 2. Data Quality and Preprocessing

### 2.1. Modulation-Cycle Phase Roll

Each modulation cycle includes the void time for its second-column separation. If an analyte's second-dimension ($^2$D) retention time ($^2t_R$) is longer than the modulation cycle, it can elute during the void time of the next $^2$D separation (or even later). No analytes otherwise are present in the void time, so only such wrap-around peaks are present in data acquired during the void time. Therefore, it is convenient both for visualization and data analysis to roll the phase of the modulation cycle back so that the all or much of data acquired during void time and any late eluting peaks that wrap into that void time appear at the top of the previous column in the image representation [9].

As seen in Figure 1, the most prominent peaks begin to elute about 1.4 s into the $^2$D separations and the tails for some of those peaks extend to later than the modulation cycle time of 2.5 s. Also, a column-bleed steak extends horizontally across the chromatogram at about 1 s into the $^2$D separations.

The phase-roll operation spirals the data to achieve the desired start position for each $^2$D chromatogram [1]. For this chromatogram, phase roll is set to 0.8 s, which leaves the bleed along the bottom of the image. Figure 2 shows the first 40 min of the terpenes-standards chromatogram after this phase roll. The peak tails are wrapped back to the tops of the image for more natural visualization of the chromatographic results.

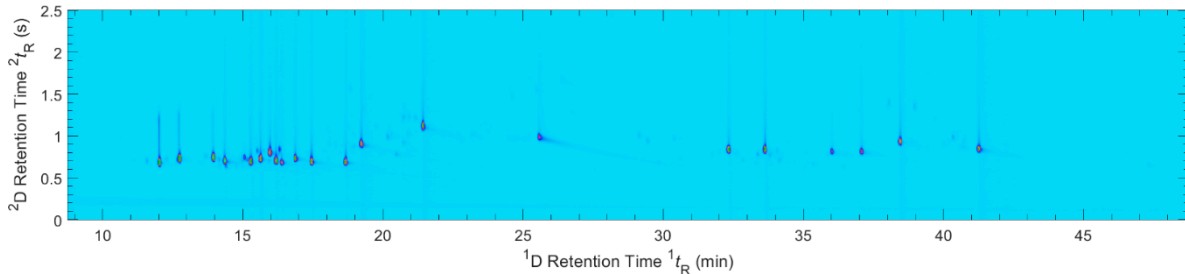

**Figure 2.** The first 40 min of the terpene-standards GCxGC chromatogram shown after −0.8 s phase roll.

### 2.2. Detector Baseline Correction

It is typical to design and build digital acquisition systems such that the detector has a non-zero baseline response even in the absence of input signal, because trying to achieve zero output risks non-detection of small signals and complicates characterizing detector noise. However, for such systems, the output values must be calibrated and corrected to account for the non-zero detector baseline.

Figure 3A indicates three horizontal lines of constant $^2$D retention times across the terpene-standards chromatogram; Figure 3B shows the detector baseline values along those lines. As can be seen, the detector baseline total intensity count (TIC) varies around a mean of about 27,000 unit-less digital number (DN). Similar (though smaller) non-zero baseline levels can be observed in the mass-spectral channels (but are not shown).

MS baseline correction builds a dynamic model of the detector baseline across each $^2$D chromatogram and in each spectral channel of the chromatogram and then subtracts it from the data [10]. Although there are parameters to control model building in GC Image software, the default settings are effective here, as they usually are. Figure 3C illustrates that the data baseline is near zero after the detector baseline correction. The variance in Figure 3C (after baseline correction) is the same as in Figure 3B (before baseline correction), although the different scale makes the variance appear relatively larger.

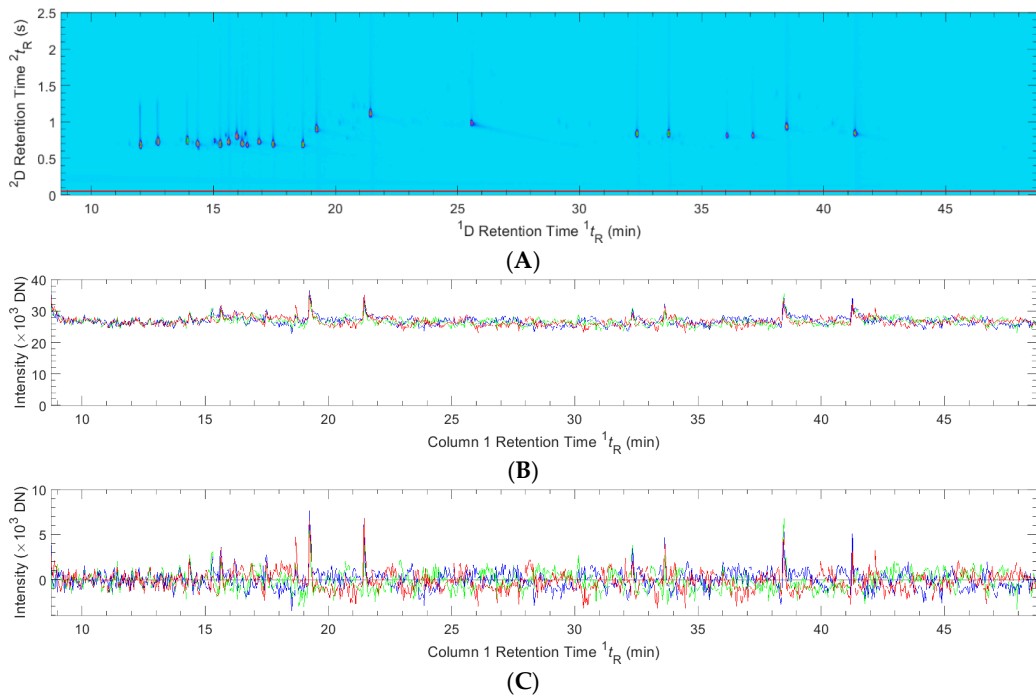

**Figure 3.** Detector baseline correction: (**A**) Horizontal lines of constant $^2$D retention times (0.45, 0.50, and 0.55 s, after phase roll); (**B**) TIC before baseline correction, with values along those lines that vary around a mean of about 27,000 DN; (**C**) TIC after baseline correction, with values that vary around zero along those lines.

## 2.3. Detector Saturation

If a detector is saturated by the intensity of the input, it cannot record a true value. For example, Figure 4 illustrates detector saturation in the center $^2$D chromatogram (plotted in green) at 15.96 min. The $^2$D chromatograms both before and after the center $^2$D chromatogram (plotted in red and blue, respectively) exhibit well-formed analyte peaks, but the center $^2$D chromatogram has a flattened peak-top, evidencing detector saturation. This effect can be observed both in the TIC (shown in Figure 4) and in the most prominent mass-spectral channels (not shown).

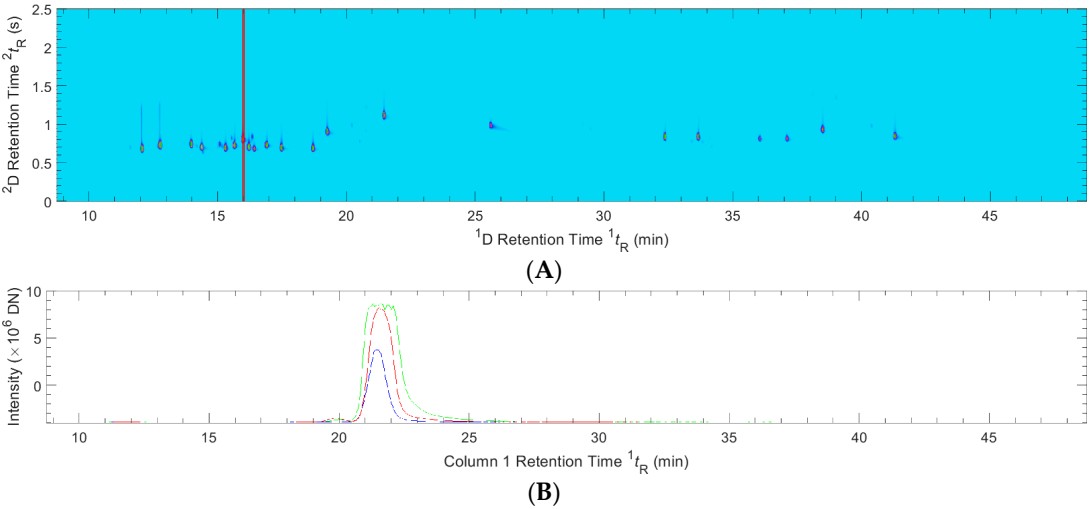

**Figure 4.** Detector saturation: (**A**) Vertical lines of constant $^1$D retention times (red 15.92, green 15.96, and blue 16.00 min); (**B**) detector TIC intensity with saturation in the middle $^2$D chromatogram, through the apex, plotted in green.

Detector saturation is problematic for peak detection algorithms, for example, possibly leading to false peak-splits and to incorrect deconvolution (especially if consistent peak shapes across $^2$D chromatograms are assumed). Another issue is that detector saturation can alter the mass spectra across the peak, potentially leading to incorrect compound identifications from MS library searches.

## 2.4. Detector Oscillation and Logarithmic Value Mapping

Detectors can impart patterned noise. The values in such patterns typically are small and may not be noticeable if linear value mapping is used for pseudocolor visualization. Logarithmic value mapping allocates more of the color scale to smaller values and so can bring into view both small peaks and small noise patterns.

For example, Figure 5 shows a small region of the terpene-standards chromatogram in which analytes are not present, pseudo-colorized with a logarithmic value scale. There is a clear oscillation in the detector TIC signal across 24 consecutive $^2$D chromatograms (The noise is present elsewhere but zooming to a small region shows the oscillation more clearly). There are approximately 18 cycles of the noise pattern over a period of 0.3 s, which closely matches the 60 Hz frequency of alternating electrical current in North America, suggesting a possible source of the oscillation. The authors have observed such pattern-noise in other data from LECO's Pegasus 4D systems.

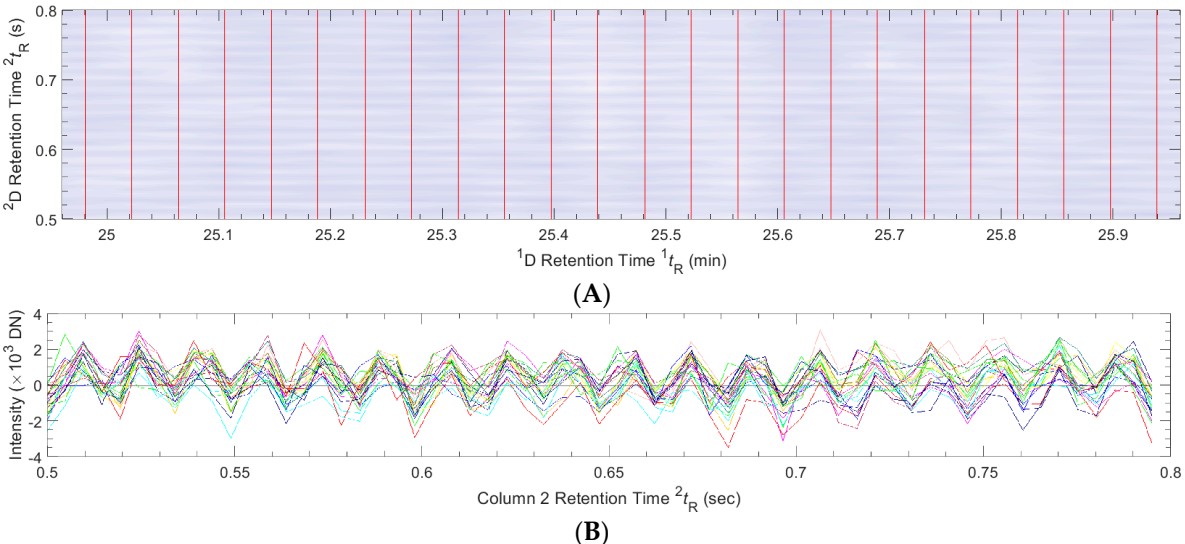

**Figure 5.** Pattern noise: (**A**) Vertical lines of constant $^1$D retention times (24.96 to 25.96 min); (**B**) pattern noise oscillations along those lines with 18 cycles from 0.5 to 0.8 s.

This oscillating noise pattern is too small to greatly affect large peaks but can pose problems for detecting and quantifying small peaks. For example, the trough of the oscillation can contribute to peak splitting and the detected quantity can increase or decrease with phase of the oscillation cycle. It is possible to attenuate the effect of such oscillations on peak detection with an adjustable 2D blur for peak-detection setting, as described in the next section.

## 3. Analyte Detection

Analytes produce 2D peaks in the retention-times plane of GC×GC data. Blob detection is the operation of detecting unimodal 2D regions containing such peaks. Here, blob/peak detection is described in four steps: Blob detection, blob filtering, true and false blob recognition, and interactive spectral review and editing. Additionally, ion-peaks detection is used to find peaks that may have been missed because they are obscured by other peaks (i.e., coelution) or noise (i.e., trace compounds).

### 3.1. Blob Detection

The Drain blob detection algorithm [11] has several parameters that allow tuning to improve performance for specific chromatograms. GC Image software has an Interactive Blob Detection tool that simplifies tuning parameter values. For the terpene-standards chromatogram, default parameters were used except:

- To deal with the detector saturation and oscillating noise described in Sections 2.3 and 2.4, the $^2$D blur parameter was set to 4.3 datapoints, which is 21.5 msec;
- To increase sensitivity for detecting faint peaks, the minimum peak threshold was set to 7 (times the estimated noise standard deviation).

With these parameter settings, 811 blobs are detected, as shown with red outlines in Figure 6A.

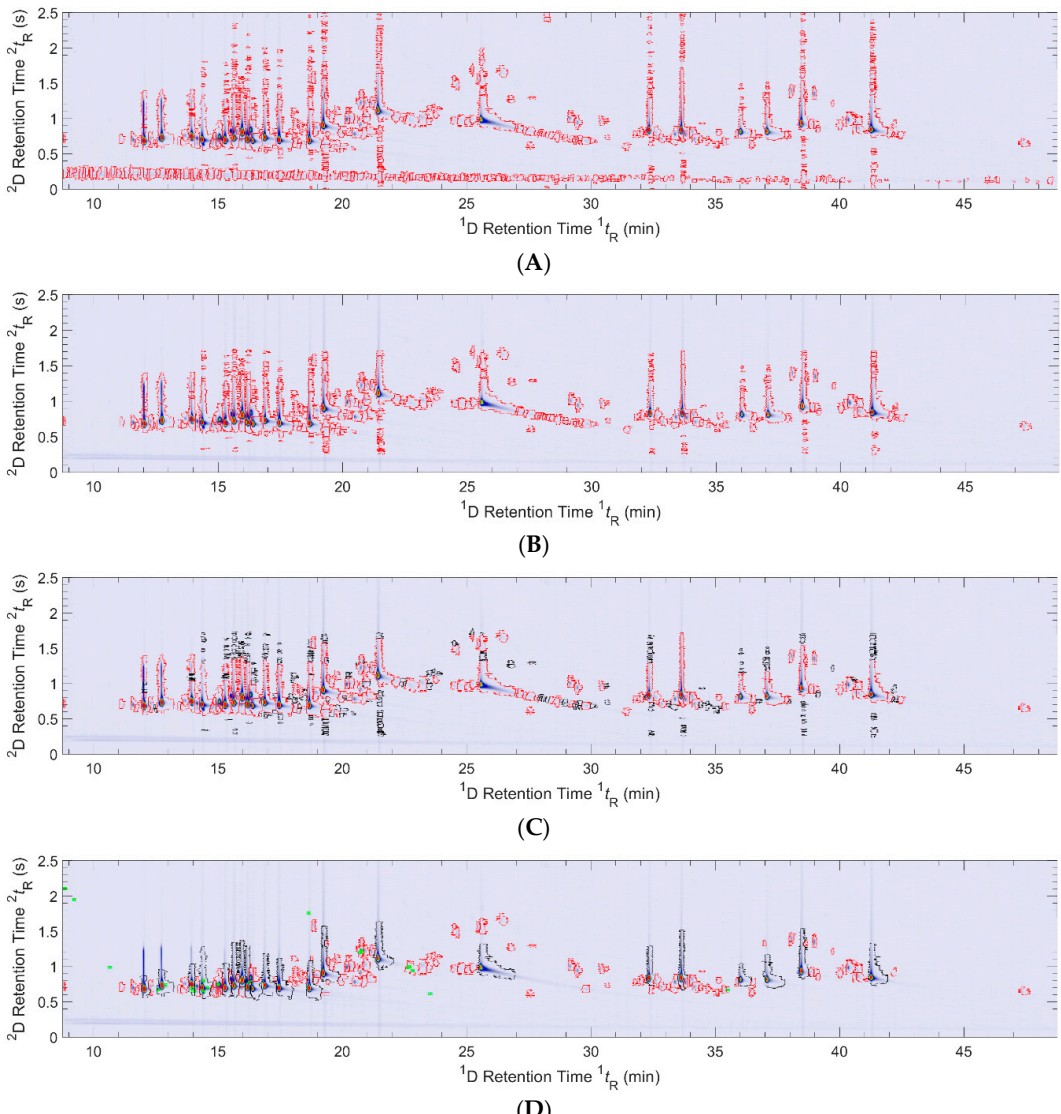

**Figure 6.** (**A**) Blob detection yielded 811 blobs (outlined in red). (**B**) Blob filtering on $^2t_R$ eliminated 398 blobs, mostly along the streak at about 0.2 s $^2t_R$, leaving 413 blobs. (**C**) Detected-blobs labeling flagged 281 blobs as false detections (outlined in black), leaving 132 blobs predicted as true detections. (**D**) QA Rapid review confirmed 148 analyte peaks, of which 21 (outlined in black) each exceed 1% of total intensity. Also, ion-peak detection yielded 20 additional peaks (shown by green apex-points) that are obscured by noise or coelution.

### 3.2. Blob Filtering

Blob filtering can be used to prevent blob detections that would violate some constraint(s). In the terpene-standards chromatogram, blobs detected during the $^2$D void time, before 0.26 s (after phase roll), are due to noise, especially the streak at about $^2t_R \approx 0.2$ s. Likewise, blobs detected after 1.71 s (after phase roll) are due to tailing, not detected analytes. Therefore, a CLIC filter [12] was used to filter blobs, requiring 0.26 s $\leq$ $^2t_R$ $\leq$ 1.71 s: "((Peak_II >= 0.26) & (Peak_II <= 1.71))". The Interactive Blob Detection tool has graphical user interfaces (GUIs), including sliders to quickly, interactively implement and parameterize such filters. Without this filter, 811 blobs are detected accounting for 97% of the chromatogram TIC; with this simple filter on $^2t_R$, 398 false blobs are filtered out leaving 413 detected blobs, shown in Figure 6B, accounting for 94% of the chromatogram TIC.

### 3.3. True and False Blob Recognition

Not all blob detections are true; for example, some result from tails, streaks, and/or noise. A promising new area for the application of machine learning is to use blob data and statistical characteristics to predict whether blob detections are true or false. With an ability to recognize false detections, aggressive blob detection is less problematic. A simple statistical model for recognizing true and false detections based on the $^2$D blob size demonstrates the potential for such an approach, which could be extended in a straightforward manner to multivariate pattern recognition models. A slider, such as in the Interactive Blob Detection tool, can be used to interactively visualize this simple model's parameter; then, setting the threshold at $\geq$0.1 s, 132 blobs are predicted true detections and 281 blobs are predicted false detections, as shown in Figure 6C, with red and black outlines, respectively. As can be seen, many of the false detections result from $^2$D tailing.

Of 115 true blobs (as determined by interactive review, described in the next section), 96 (83%) were correctly predicted true and 19 (17%) were incorrectly predicted false. Of 259 false blobs, 29 (11%) were incorrectly predicted true and 230 (89%) were correctly predicted false. So, the overall accuracy was better than 87% (326/374). Figure 7 shows plots of Precision versus Recall and the Receiver Operating Characteristic (ROC) for this predictor. As can be seen, the 83% true-positive-rate performance (with the threshold at 0.1 s) is a fair balance between false positives and false negatives, but lower rates of false negatives could be achieved (at the cost of additional false positives). During interactive review 32 of the 413 detected blobs required merging and 7 of the 413 detected blobs required splitting. Of the 32 detected blobs that required subsequent merging, only one was predicted to be a true detection; of the 7 detected blobs that required subsequent splitting, six were detected as true detections. This simple model indicates there is great potential for multivariate, multiclass, multisample pattern recognition for labelling blob detections and machine learning for this operation is an area of our ongoing research.

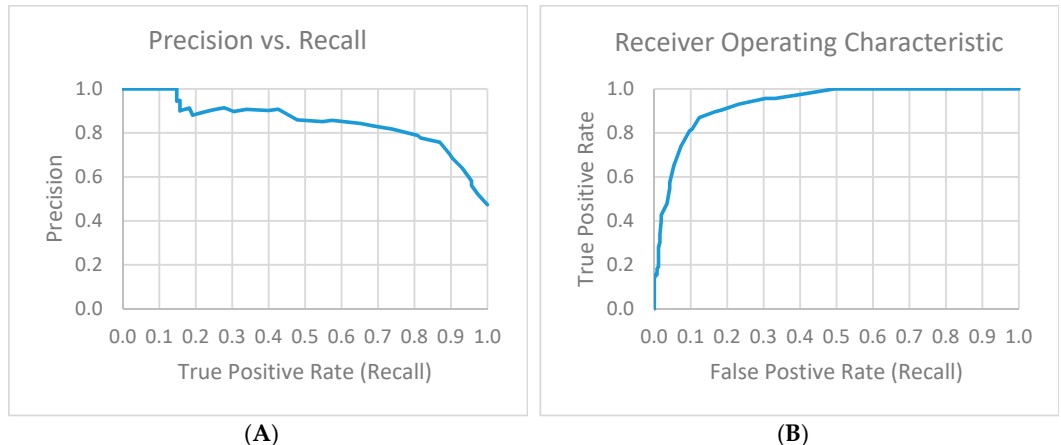

**Figure 7.** Predicting true and false analyte detections based on $^2$D blob size. (**A**) Precision-Recall graph; (**B**) ROC graph.

### 3.4. Interactive Blob Review and Editing

Predictions of true and false blob detections should be reviewed for correctness. The Spectral Review mode of the GC Image QA Rapid Screen tool facilitates blob review and editing. The tool shows simultaneously the blob table, a thumbnail of the entire chromatogram and zoomed TIC and selected ion count (SIC) images, 1D intensity profile(s), and the mass spectrum for the selected blob. When a blob is selected in the blob table, the TIC and SIC images, 1D intensity profiles, and mass spectrum are updated dynamically for the selected blob. With this tool, it is possible to scan quickly through each blob. The interactive review can be especially fast if the predictions of true and false blob detections are highly accurate.

Interactive review confirmed 96 correct true-blob predictions and corrected 29 incorrect false-blob predictions. In addition, 32 detected blobs were judged to result from incorrectly split peaks and 7 detected blobs were judged to result from either incorrectly merged peaks of a peak merged with a streak or tail. Generally, the incorrectly split blobs had low intensity, so the splits could have been related to detector oscillation (described earlier), and the incorrectly merged blobs were in tight proximity. GC Image software has a blob editing tool that supports simple point-and-click merging of incorrectly split blobs and splitting of incorrectly merged blobs. So, after interactive review there are 144 analyte blobs (96 confirmed true-blob predictions, 29 corrected false-blob predictions, 16 blobs after merging split blobs, and 3 new blobs from splits).

Of the 144 analyte blobs after QA Rapid Screen, 21 have volume (summed intensity) that exceeds 1% of the total percent response of all blobs and together account for more than 94% of the total blob response. In Figure 6D, the Top 21 (i.e., largest TIC response) analyte blobs are outlined in black and the other 123 are outlined in red. Table 1 lists the Top 21 analytes with Blob ID, TIC percent response (relative to the total of all blobs), retention times, and spectrum base peak. The other attributes in Table 1 are described in Section 4. The full table, with all 144 detected analytes after $I^T$ calibration and identification, is included in the Supplementary Materials as Table S1.

### 3.5. Ion-Peaks Detection

Peaks also can be detected in individual ion channels and then can be collected as sets of coincident ion-peaks. For the terpene-standards chromatogram, ion-peaks detection was parameterized in GC Image software with default values except intensity threshold 50, peak-width threshold 0.075 s (15 datapoints), peak SNR threshold 5, and ion-sampling interval 2. With these settings, 960 ion-peak sets were detected. Less restrictive settings, e.g., a smaller peak-width threshold would detect even more trace compounds, but relaxing detection constraints also increases the work required for confirmation.

Of the 960 ion-peak sets detected with these settings, 680 are detected along the horizontal streak in the void time and 62 are detected along the tail of the large peak eluting at 25.54 min, so are deleted (with selection by drawing a polygon around them), leaving 228 ion-peak sets. By interactive screening with QA Rapid Review, of those remaining, all but 20 were consistent with previously detected analytes (i.e., redundant) or due to noise. Of the remaining 20, 6 were judged new detections (i.e., too faint for the blob detection settings) and 14 were judged coeluted with previously detected analytes. The locations of these ion-peaks are shown in Figure 6D with green apex-points. A table of these 20 ion peaks is included in the Supplementary Materials as Table S2.

**Table 1.** Analyte ID, compound identification by #1 hit of MS search, blob metadata (percent response, $^1$D and $^2$D retention times, computed $I^T$, base peak), library search result for #1 hit (direct match factor, reverse match factor, probability, $I$, base peak), and identification test results (direct match factor > 700, analyte base peak = #1 hit base peak, Analyte $I^T$ − Library $I \leq 15$).

| | Library Search #1 Hit | | Blob | | | | | Library | | | | | Tests | | |
|---|---|---|---|---|---|---|---|---|---|---|---|---|---|---|---|
| ID | Compound Name | CAS# | %Rsp | $^1t_R$ | $^2t_R$ | $I^T$ | BPk | DMF | RMF | Prob | $I$ | BPk | DMF | BPk | RI |
| 1 | ß-Myrcene | 123-35-3 | 5.38 | 14.33 | 0.70 | 983 | 41 | 907 | 907 | 56 | 983 | 41 | PASS | PASS | PASS |
| 2 | D-Limonene | 5989-27-5 | 5.59 | 16.17 | 0.70 | 1028 | 68 | 890 | 890 | 15 | | 68 | PASS | PASS | |
| 3 | Linalool | 78-70-6 | 4.99 | 19.21 | 0.92 | 1086 | 43 | 929 | 929 | 73 | 1086 | 71 | PASS | FAIL | PASS |
| 4 | ß-Pinene | 127-91-3 | 5.41 | 13.92 | 0.75 | 973 | 93 | 944 | 945 | 50 | 973 | 93 | PASS | PASS | PASS |
| 5 | Camphene | 79-92-5 | 5.86 | 12.75 | 0.72 | 946 | 93 | 947 | 962 | 38 | 946 | 93 | PASS | PASS | PASS |
| 6 | (1R,2R,5S)-5-Methyl-2-(prop-1-en-2-yl)cyclohexanol | 29141-10-4 | 4.39 | 21.42 | 1.13 | 1139 | 41 | 930 | 931 | 45 | | 69 | PASS | FAIL | |
| 7 | Humulene | 6753-98-6 | 4.95 | 33.63 | 0.84 | 1451 | 93 | 886 | 888 | 49 | 1451 | 93 | PASS | PASS | PASS |
| 8 | α-Pinene | 80-56-8 | 5.13 | 15.29 | 0.70 | 1003 | 93 | 921 | 929 | 16 | 933 | 93 | PASS | PASS | FAIL |
| 9 | α-Pinene | 80-56-8 | 5.30 | 12.00 | 0.70 | 933 | 93 | 945 | 952 | 33 | 933 | 93 | PASS | PASS | PASS |
| 10 | Caryophyllene | 87-44-5 | 4.52 | 32.33 | 0.84 | 1419 | 41 | 926 | 926 | 36 | 1419 | 93 | PASS | FAIL | PASS |
| 11 | Cyclohexene, 1-methyl-4-(1-methylethylidene)- | 586-62-9 | 4.78 | 18.67 | 0.68 | 1079 | 93 | 916 | 923 | 15 | 1079 | 93 | PASS | PASS | PASS |
| 12 | τ-Terpinene | 99-85-4 | 5.43 | 17.46 | 0.69 | 1050 | 93 | 876 | 878 | 12 | 1050 | 93 | PASS | PASS | PASS |
| 13 | ß-Ocimene | 13877-91-3 | 4.31 | 16.88 | 0.73 | 1037 | 93 | 938 | 938 | 35 | 1037 | 93 | PASS | PASS | PASS |
| 14 | 1,3-Cyclohexadiene, 1-methyl-4-(1-methylethyl)- | 99-86-5 | 4.93 | 15.63 | 0.74 | 1010 | 93 | 907 | 916 | 19 | 1010 | 121 | PASS | FAIL | PASS |
| 15 | α-Bisabolol | 515-69-5 | 4.49 | 41.25 | 0.85 | 1668 | 43 | 929 | 944 | 75 | 1668 | 43 | PASS | PASS | PASS |
| 16 | o-Cymene | 527-84-4 | 5.02 | 15.96 | 0.79 | 1025 | 119 | 944 | 958 | 63 | 1025 | 119 | PASS | PASS | PASS |
| 17 | 5-Azulenemethanol, 1,2,3,4,5,6,7,8-octahydro-α,α,3,8-tetramethyl- | 13822-35-0 | 4.10 | 38.42 | 0.94 | 1588 | 59 | 893 | 904 | 30 | | 59 | PASS | PASS | |
| 18 | α-Pinene | 80-56-8 | 2.52 | 16.38 | 0.69 | 1030 | 93 | 913 | 920 | 19 | 933 | 93 | PASS | PASS | FAIL |
| 19 | 1,6,10-Dodecatrien-3-ol, 3,7,11-trimethyl-, (E)- | 40716-66-3 | 2.43 | 37.04 | 0.82 | 1549 | 41 | 923 | 925 | 39 | 1549 | 69 | PASS | FAIL | PASS |
| 20 | Geraniol | 106-24-1 | 3.19 | 25.54 | 0.99 | 1237 | 41 | 922 | 922 | 67 | 1237 | 69 | PASS | FAIL | PASS |
| 21 | 1,6,10-Dodecatrien-3-ol, 3,7,11-trimethyl- | 7212-44-4 | 1.46 | 36.00 | 0.82 | 1519 | 41 | 939 | 953 | 61 | 1551 | 41 | PASS | PASS | FAIL |

## 4. Analyte Identification

### 4.1. MS Search Optimization

A series of experiments aimed to maximize performance of MS library search with respect to several methods and settings for extracting mass spectra from the chromatogram and several options for the MS library presearch. The results of these experiments are of interest for this case study, but the ultimate goal is to develop a new method for automating the determination of settings for MS library search for a given chromatogram (or set of chromatograms) without a priori identifications.

#### 4.1.1. Experimental Variables

For these experiments, MS library search was conducted with the 2017 NIST/EPA/NIH Mass Spectral Library with Search Program (NIST 17) using the Main Library (mainlib) database [13]. The experimental variables (i.e., method parameters for optimization) are listed in Table 2. The first variable is an option for the NIST 17 MS Search Program. The other four variables are options for extracting a mass spectrum for each analyte. As will be shown, the options for extracting spectra are most impactful on performance.

**Table 2.** Experimental variables for optimizing MS search performance.

| Abbr. | Description | Options |
|---|---|---|
| Search | NIST Identity Search Presearch | [None, Quick, Normal] |
| Source | Source of Mass Spectrum | [Apex, Blob] |
| Integr. | Percent of Apex for Integration (Blob Source Type only) | [Number 0–100] |
| Subtr. | Background Subtraction | [None, Start, Start & End] |
| Thresh. | MS Peak Intensity Threshold | [Non-negative number] |

For the NIST Search options, Identity Search can be configured to employ presearch to select library spectra for matching, either: *None*, which matches the search spectrum with all library spectra; *Quick*, which uses peak-scaling screening of the spectrum's eight most intense peaks to select library spectra for matching; or *Normal*, which uses peak-scaling screening and three additional screens to select library spectra for matching. For details of these screening methods, see the NIST MS Spectral Search Program User's Guide [14].

For the spectrum extraction options, the Source spectrum can be taken either from the *Apex*, i.e., the single spectrum with the largest TIC within the blob, or from the *Blob*, by summing multiple spectra from within the blob. In computing a Blob spectrum, the individual spectra within the blob that are Integrated (i.e., summed) can be limited to those that are most intense by setting a threshold on the per-spectrum TIC as a percentage of the apex TIC. Chromatographic background can be removed from the Apex spectrum or from each spectrum summed in the Blob spectrum by Subtraction of a background spectrum. The background spectrum can be taken either just before the *Start* of the blob in the $^2$D chromatogram containing the spectrum or as the average at the *Start & End*, i.e., just before the start and just after the end of the blob in the $^2$D chromatogram. Finally, the smallest intensity ion peaks can be regarded as noise and removed from the search spectrum, either Apex or Blob, by setting the intensity threshold on the TIC.

The dependent variables for evaluating MS Search performance are listed in Table 3. In Table 3, "All" refers to all 144 analytes in the chromatogram and "Top" refers to the set of 21 analytes that had greater than 1% TIC response relative to the sum for all 144 analytes. Several of the metrics are the simple averages of values from the highest ranked (#1) hits returned by NIST MS search [14] over the sets of All and Top analytes: direct match factor (DMF), reverse match factor (RMF), and probability value (Prob.). The averages of direct and reverse match factors (AMF) for the #1 hits also are reported. The final metric, the coefficient of determination ($R^2$), is based on the library retention-indices for the

#1 hits, available for many compounds in NIST 17, and is described in the next paragraph. Two of the most informative values are *AMF All* and $R^2$ *All* (listed in shaded rows).

**Table 3.** Dependent variables for evaluating MS search performance with direct, reverse, and average match factors (DMF, RMF, AMF), probability (Prob.), and coefficient of determination ($R^2$) for All and Top analytes.

| Abbr. | Description | Range |
|---|---|---|
| DMF All | Direct Match Factor for #1 hit each analyte, averaged for all analytes | [0–999] |
| DMF Top | Direct Match Factor for #1 hit each analyte, averaged for top analytes | [0–999] |
| RMF All | Reverse Match Factor for #1 hit each analyte, averaged for all analytes | [0–999] |
| RMF Top | Reverse Match Factor for #1 hit each analyte, averaged for top analytes | [0–999] |
| Prob. All | Probability for #1 hit each analyte, averaged for all analytes | [0–100] |
| Prob. Top | Probability for #1 hit each analyte, averaged for top analytes | [0–100] |
| AMF All | Average of DMF & RMF for #1 hit each analyte, averaged for all analytes | [0–999] |
| AMF Top | Average of DMF & RMF for #1 hit each analyte, averaged for top analytes | [0–999] |
| $R^2$ All | $R^2$ for Linear Fit with ($^1t_R$, $I$) for #1 hits of all analytes | [0–1] |
| $R^2$ Top | $R^2$ for Linear Fit with ($^1t_R$, $I$)) for #1 hits of top analytes | [0–1] |

As shown in Figure 8, for each analyte, the $^1t_R$ and library $I$ for the #1 hit can define the abscissa and ordinate for a point in an (x,y) plot. If the library record for the #1 hit does not have a value for $I$, the analyte is not plotted. Most of the points for this example (and others described below) fall along the regression line fit to these points, but those some distance off the line are indicative of misidentified analytes, i.e., the analyte $^1t_R$ time and library $I$ for the #1 hit are inconsistent. (Other chromatograms might require fitting by a non-linear model.) Thus, $R^2$ for the model provides an evaluation of misidentified analytes. For example, linear regression fits the top analytes (Figure 8B, on the right) much better than all analytes (Figure 8A, on the left), with $R^2$ of 0.9862 and 0.7804 respectively. The better fit indicates that the identifications of top analytes are better, which is as expected. It would be ideal to know the identities for the 144 compounds, but such knowledge is lacking for this chromatogram, as it is for most analytes in complex analyses, and would be lacking for a general method for optimizing MS Search settings for a given chromatogram.

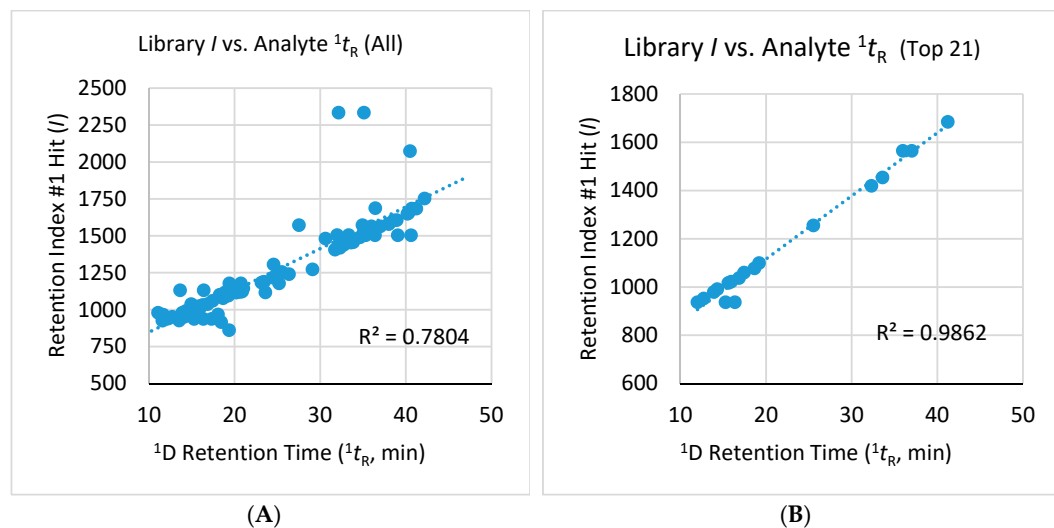

**Figure 8.** Library $I$ versus analyte $^1t_R$ with linear regression fit for NIST Normal Search with Peak Source MS and no Integration, Subtraction, or Thresholding (Row 1 of Table 4): (**A**) All analytes with $I$ for the #1 hit; (**B**) Top 21 analytes with $I$ for the #1 hit.

### 4.1.2. Experimental Results

Table 4 lists the conditions and results for both sets of analytes (All and Top) for a few of the experiments. A table with all experiments is included in the Supplementary Materials as Table S3. The following discussion refers to rows of that table. For example, Rows 1–3 show the results for different NIST Search settings (None, Quick, and Normal), all with Apex spectra and no Integration, Subtraction, or Thresholding. For these cases, there is very little difference in the results. For the top analytes, the average DMF, RMF, and $R^2$ are identical (923, 928, and 0.99, respectively). For all analytes, no presearch yields a slightly better average DMF (775), AMF (797), and $R^2$ (0.80) than Normal and Quick presearch (DMF of 774 and 772, AMF of 796 for both, and $R^2$ of 0.78 and 0.75). However, it is important to note that computational time is a significant consideration in this setting. Both Quick and Normal searches are quite fast, for this search, about 10 s and 15 s, respectively, but searching with None for Presearch required about 900 s. Based on these results (and similar results for Blob spectra on Rows 4–6), Normal search provides some improvement over Quick search and has much faster computation with nearly the same performance as no presearch. Therefore, for these experiments to maximize MS search performance, Normal search is used to more quickly converge to optimal or near-optimal settings.

**Table 4.** Mass spectral search experimental settings (source spectrum, blob spectrum integration threshold, spectrum subtraction, intensity threshold, presearch) and resulting performance for All analytes and Top analytes (direct match factor, reverse match factor, probability, average match factor, coefficient of determination). Groups of experiments are shown with alternate shading. Results for All analytes are colorized by column to indicate performance (red for low, green for high).

| | Experimental Settings | | | | | Results All | | | | | Results Top | | | | |
|---|---|---|---|---|---|---|---|---|---|---|---|---|---|---|---|
| Row | Src. | Intgr. | Subtr. | Thr. | Search | DMF | RMF | Prob. | AMF | $R^2$ | DMF | RMF | Prob. | AMF | $R^2$ |
| 1 | Apex | 0 | None | 0 | None | 775 | 818 | 21% | 797 | 0.80 | 923 | 928 | 39% | 926 | 0.99 |
| 2 | Apex | 0 | None | 0 | Quick | 772 | 820 | 25% | 796 | 0.75 | 923 | 928 | 40% | 926 | 0.99 |
| 3 | Apex | 0 | None | 0 | Norm | 774 | 818 | 25% | 796 | 0.78 | 923 | 928 | 40% | 926 | 0.99 |
| 4 | Blob | 0 | None | 0 | None | 785 | 835 | 18% | 810 | 0.72 | 914 | 922 | 38% | 918 | 0.99 |
| 5 | Blob | 0 | None | 0 | Quick | 783 | 834 | 23% | 809 | 0.68 | 914 | 922 | 39% | 918 | 0.99 |
| 6 | Blob | 0 | None | 0 | Norm | 784 | 835 | 23% | 809 | 0.72 | 914 | 922 | 40% | 918 | 0.99 |
| 21 | Apex | 0 | None | 27 | Norm | 791 | 831 | 27% | 811 | 0.92 | 923 | 928 | 40% | 926 | 0.99 |
| 43 | Blob | 0 | None | 700 | Norm | 808 | 840 | 26% | 824 | 0.86 | 914 | 922 | 40% | 918 | 0.99 |
| 51 | Apex | 0 | Start | 0 | Norm | 759 | 803 | 25% | 781 | 0.69 | 924 | 928 | 41% | 926 | 0.99 |
| 61 | Blob | 0 | Start | 0 | Norm | 753 | 795 | 24% | 774 | 0.66 | 913 | 920 | 39% | 917 | 0.99 |
| 72 | Apex | 0 | S&E | 0 | Norm | 761 | 801 | 25% | 781 | 0.57 | 923 | 929 | 40% | 926 | 0.99 |
| 82 | Blob | 0 | S&E | 0 | Norm | 765 | 814 | 24% | 789 | 0.70 | 915 | 921 | 39% | 918 | 0.99 |
| 154 | Blob | 45 | None | 250 | Norm | 826 | 851 | 30% | 838 | 0.95 | 920 | 926 | 40% | 923 | 0.99 |

Results for different NIST Search settings with Blob spectra and no Integration, Subtraction, or Thresholding are in Rows 4–6. Comparisons to the results for Apex spectra (in Rows 1–3) are mixed. The average AMF for All analytes is better with Blob spectra, but the average AMF for the Top analytes is better with Apex spectra. Average $R^2$ for All analytes is better with Apex spectra and average $R^2$ for Top analytes is about the same with Apex or Blob spectra. The comparative results for Apex and Blob spectra are more interesting when each is optimized with respect to other settings, as described below.

One of the most configurable and effective settings for extracting the source spectra is the spectral intensity Threshold, which can be set to remove the smallest intensity (and typically noisiest) spectral peaks. Figure 9 shows that as the Threshold is increased from 0, both AMF and $R^2$ for All analytes improve, but then after an "optimal" threshold value is reached, performance begins to decrease. For the Apex spectra (Figure 9A), the performance peaks with threshold about 25; for the Blob spectra (Figure 9B), which have greater summed intensities than the Apex spectra, the performance peaks with threshold about 700. The maximum performance of AMF All is about 811 for the Apex spectra (Row 21) and about 824 for the Blob spectra; the maximum $R^2$ All is about 0.92 for Apex spectra and

about 0.90 for Blob spectra. Both metrics are somewhat noisy, which means the optimal settings for peak performance cannot be determined exactly, but the best AMF All with Blob spectra is better than with Apex spectra and the best $R^2$ All performance is about the same for both sources. Full results for these experimental results are shown on Rows 3 and 7–27 for Apex spectra and Rows 6 and 28–50 for Blob spectra.

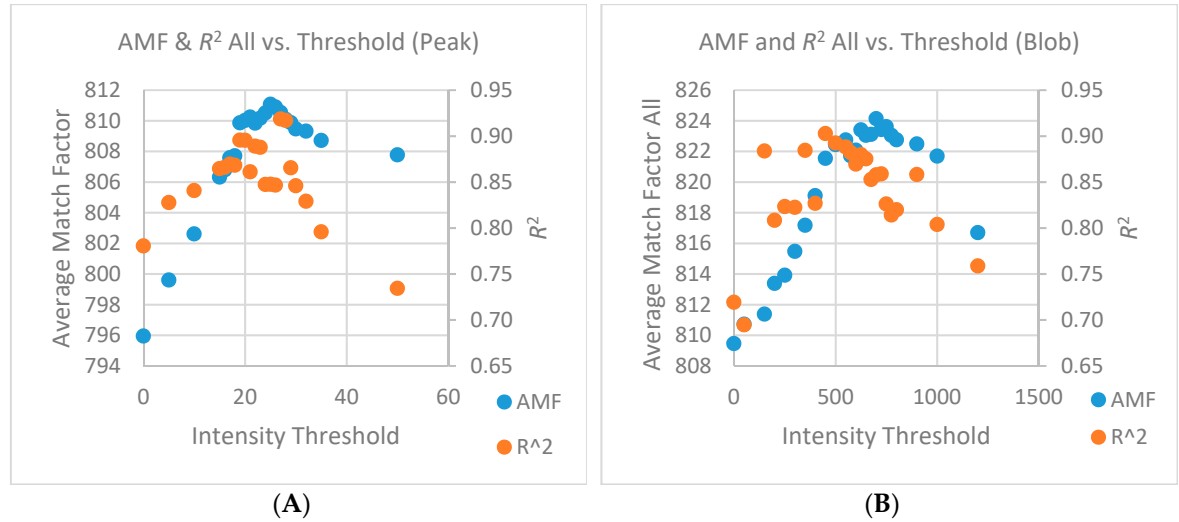

**Figure 9.** AMF and $R^2$ All performance with intensity thresholding for (**A**) Peak spectra and (**B**) Blob spectra.

Another method for noise suppression is to subtract background spectra; either the spectrum immediately before the blob or the average of the spectra immediately before and after the blob. With Start spectrum subtraction, Apex spectra yield AMF All of 781 and $R^2$ All of 0.69 (Row 51) and Blob spectra yield AMF All of 774 and $R^2$ All of 0.66 (Row 61). These are below the marks achieved without Start subtraction (for Apex and Blob spectra respectively, AMF All of 796 and 809 and $R^2$ All of 0.78 and 0.72, on Rows 3 and 6). Combining intensity Thresholding with Start subtraction can increase performance, but even so, AMF All never rises above 800 for either Apex or Blob spectra (Rows 51–71). Subtracting the average of the Start and End spectra performs slightly better, with 781 and 0.57 for Apex spectra (Row 72) and 787 and 0.65 for Blob spectra (Row 82). Combining intensity Thresholding with Start & End subtraction can increase performance, but even so, AMF All never rises above 798 for Apex spectra nor above 807 for Blob spectra (Rows 72–92). By comparison, Intensity Thresholding of Blob spectra without background subtraction reaches 824 for AMF All (Row 43).

A final method of extracting a spectrum is to sum only the larger spectra (which have greater SNR) within a blob. Here, the region for Integration is expressed as a threshold percentage for each spectrum's TIC relative to the apex spectrum TIC. Simultaneously optimizing both Integration limits and intensity Thresholding requires exploration of a 2D parameter space. Figure 10 illustrates the effect of changing the Integration region for four different levels of intensity Thresholding. (Many levels of intensity Thresholding are included in Rows 93–184, but only these four cases, for which performance is best, are plotted.) As expected, as the Integration region is made smaller (with a larger percentage parameter), the optimal intensity Threshold also decreases.

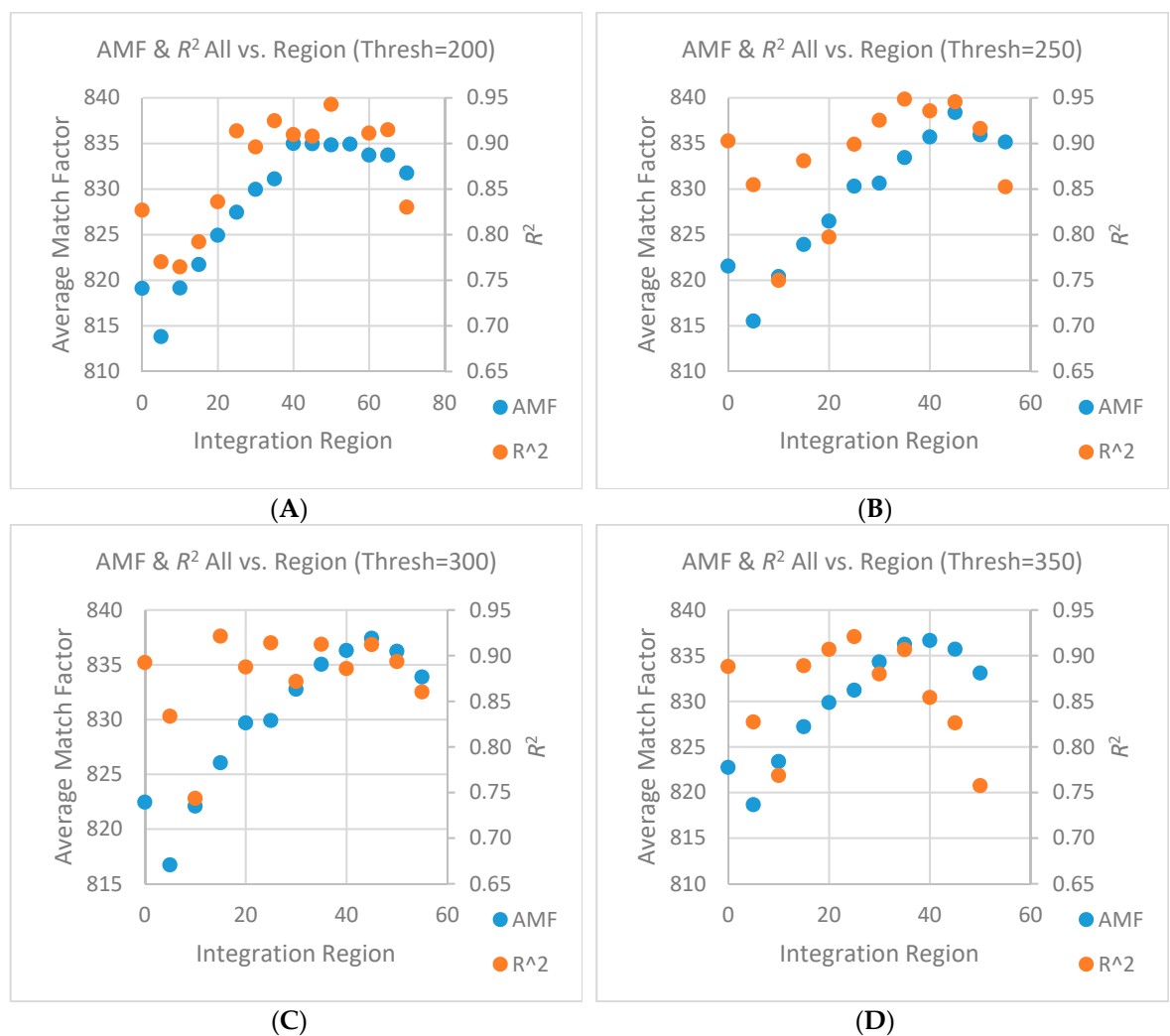

**Figure 10.** AMF and $R^2$ All performance with Integration constraint for four levels of intensity Thresholding: (**A**) 200; (**B**) 250; (**C**) 300; (**D**) 350.

### 4.1.3. Maximum Performance

In these experiments, the MS search performance is maximized with the Integration threshold 45% and intensity Threshold 250 (Figure 10C; Row 154), at which point AMF All exceeds 838 and $R^2$ All is about 0.95. Figure 11 plots the library *I* versus analyte $^1t_R$ for these settings that maximize performance. This performance compares to 796 and 0.78 for the simple Apex spectrum and 809 and 0.72 for the simple summed Blob spectrum. The comparative performance is illustrated with column bars in Figure 12. Tuning the extraction parameters significantly improves performance.

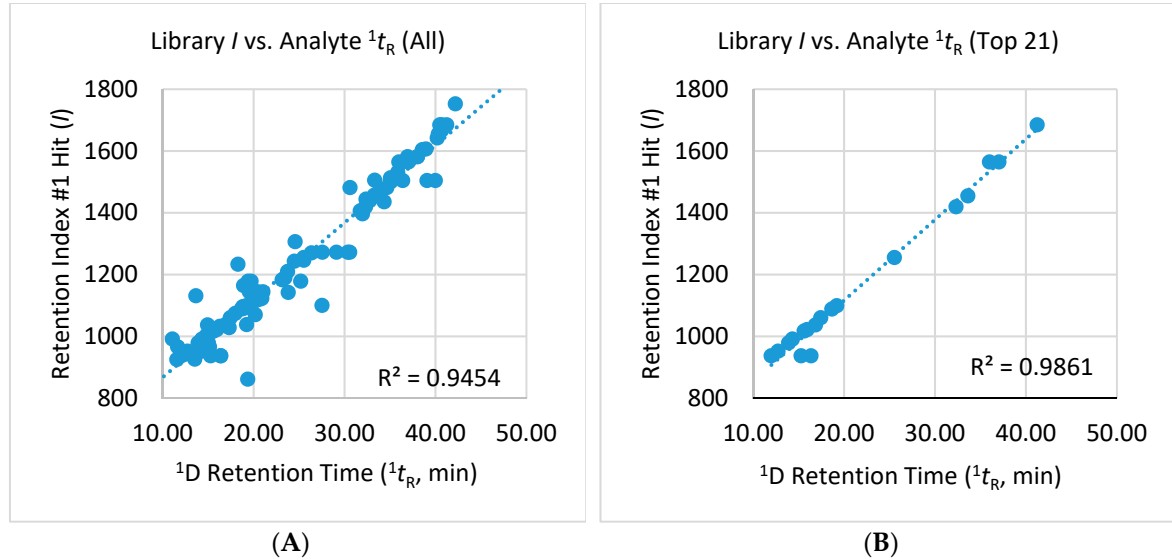

**Figure 11.** Library *I* versus analyte $^1t_R$ with linear regression fit for Blob spectrum with integration 45%, no background subtraction, and intensity threshold 250 (Row X of Table 4): (**A**) All analytes; (**B**) Top 21 analytes.

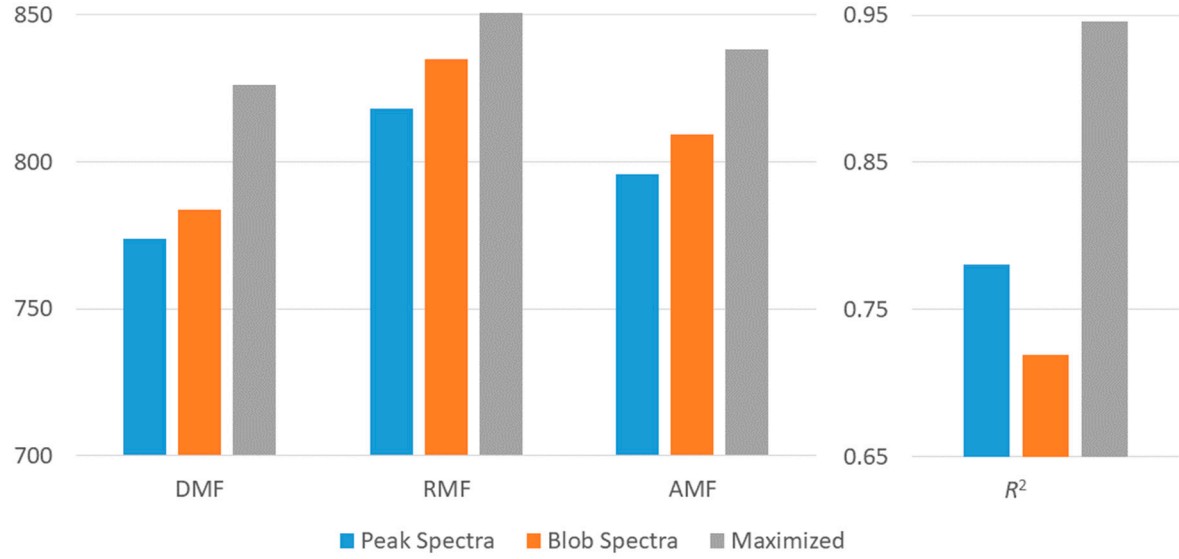

**Figure 12.** Comparative performance with peak spectra, blob spectra and maximized extraction settings.

*4.2. MS Search*

MS search was performed using the optimized search settings as described in Section 4.1 with the NIST 17 mainlib. Table 1 lists the search results for the Top 21 analytes with:

1.  The library compound name and CAS number for the MS search #1 hit for each analyte, i.e., the preliminary identification, subject to confirmation by the analyst;
2.  The library search results for the #1 hit, including DMF, RMF, probability, *I*, and base peak.

The average AMF for all search spectra is 838, but the AMF for individual analytes varies widely and, as illustrated in Figure 13, is related to percent response. Because there is such a large dynamic range for percent response, that axis is shown with a log scale. As expected, the larger intensity analytes have better AMF. The AMF for analytes with greater than 0.1% response (N = 32) ranges from 873–955 and averages 917. For analytes with 0.02% to 0.1% response (N = 54), the AMF ranges from 749–951

and averages 849. For analytes with less than 0.02% response (N = 58), the AMF ranges from 564–877 and averages 785.

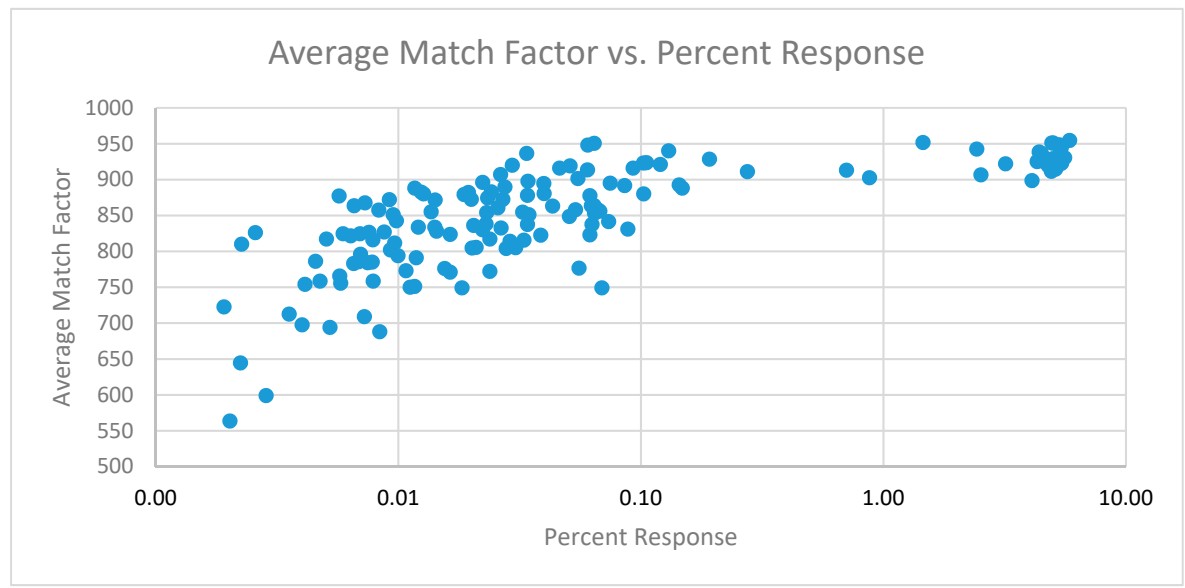

**Figure 13.** Average match factor as a function of percent response.

Despite high match factors, there clearly are identification errors with the #1 hits. For example, as seen in Table 1, Analytes 8, 9, and 18 all have the same #1 hit, $\alpha$-Pinene. In such cases, automated tests can be helpful for screening for misidentifications. One useful test is whether the analyte $^1 t_R$ is consistent with the library $I$ of the #1 hit. As described in the next section, preliminary $I^T$ calibration can be performed quickly, even without calibration standards or known compounds, using the library retention indices from preliminary identifications.

### 4.3. Retention Index Calibration

For authoritative $I^T$ calibration, analysts should experimentally measure retention times for reference compounds; linear temperature programming is preferred [15]. However, ad hoc $I^T$ calibration can be computed automatically from putatively identified compounds to produce a consistent $I^T$ model that can be used effectively to check preliminary compound identifications.

This case study develops a method for preliminary $I^T$ calibration based on fitting the calibration model to the observed relationship between $^1 t_R$ and the #1 hit $I$. Figure 14A plots ($^1 t_R$, $I$) for the #1 hit for the 21 Top analytes, with piecewise linear interpolation. Three of the 21 Top analytes (IDs 2, 6, and 17) are not plotted because the library did not have $I$ for the #1 hit. In Figure 14B, three additional analytes (IDs 8, 18, and 21) are eliminated because the retention indices of the #1 hit were inconsistent with the others, leaving 15 analytes for $I^T$ calibration. All of those eliminated were identified as the same compound as another analyte in the Top 21 (indicative of misidentifications). The result (in Figure 14B) is a credible model for preliminary $I^T$ calibration, even without a priori known standards. The analyte $I^T$ values in Table 1 are computed based on the piecewise linear $I^T$ calibration function shown in Figure 14B. Of course, if preliminary identifications are replaced by more authoritative identifications, the $I^T$ calibration can be recomputed.

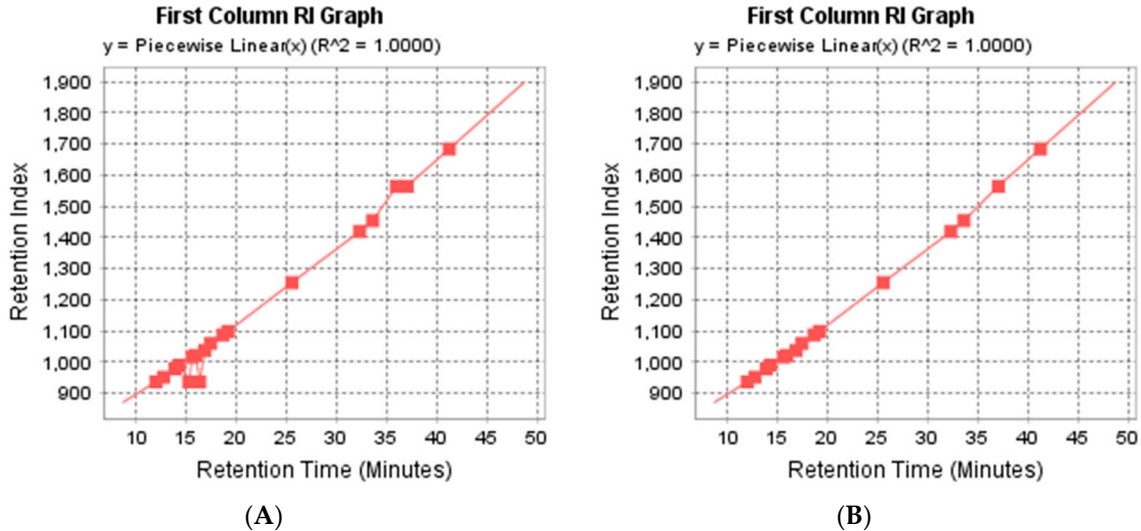

**Figure 14.** Library I of #1 hits versus analyte $^1t_R$: (**A**) 18 of Top 21 analytes (3 #1 hits lacked library I); (**B**) 15 of Top 21 analytes consistent with piecewise linear $I^T$ calibration model.

Here, library search was with Column Type set to "Semi-standard Non-Polar (e.g., DB5)", but results using "Standard Non-Polar (e.g., DB-1)" yielded similar results for checking compound identifications (presented in the next section), because retention indices are computed relatively and the model is locally linear so the windows for the RI checks are relatively about the same.

*4.4. Expressions for Checking Compound Identifications*

Checks of preliminary compound identifications can be implemented in computable expressions, e.g., using CLIC [12]. Here, three such identification checks are employed as examples.

1. DMF. This check requires that the DMF of the analyte spectrum with the library compound spectrum exceed a specified threshold, e.g.,:IF(Library_Match_Factor >= 850,"PASS","FAIL")
2. Base Peak (BPk). This check requires that the base peak of the analyte spectrum is the same as the base peak of the library compound:IF(MASSRANK(1) = Library_Base_Peak,"PASS","FAIL");
3. RI. This check requires that the difference between the computed analyte $I^T$ and the $I$ of the library compound is not greater than a specified threshold, e.g.,: IF(ABS(LRI_I-LibraryRI) < 15,"PASS","FAIL") This tolerance is somewhat large, but is justified by the preliminary purpose, lack of details about the chromatographic conditions, and pressure drop across the $^1$D column due to outlet restriction generated by the modulator.

The results of these checks for each of the Top 21 analytes are listed in Table 1. Additional checks or combinations of checks similarly could be employed.

Referencing Table S1 in the Supplementary Materials, analytes with greater than 0.1% response (N = 32, including the Top 21 analytes), 100% passed the DMF check. As expected, the passing rate decreases with analyte intensity. For analytes with 0.02% to 0.1% response (N = 54), the DMF passing rate is 56%; for analytes with less than 0.02% response (N = 58), the DMF passing rate falls to 16%.

A failed RI test is highly indicative of misidentification. Of the Top 21 analytes, shown in Table 1, three identifications fail the RI check—the same three analytes eliminated from the preliminary $I^T$ calibration. Two of those (Analytes 8 and 18) are identified as α-Pinene, whereas Analyte 9, also identified as α-Pinene, passes all three tests.

1. For Analyte 8, the NIST 17 Replicates Library (replib) has a good match for "3-Carene" aka Δ-3-Carene (DMF = 905, RMF = 924), which has a passing $I$ = 1011 (compared to 1010 for the computed $I^T$).

2. For Analyte 18, the #2 hit with mainlib (DMF = 906, RMF = 907) is "β-Ocimene" (unspecified isomer), which has a passing $I$ = 1037 (compared to 1029 for the computed $I^T$). However, the preliminary identification of Analyte 13 was "β-Ocimene". Further examination suggests Analyte 13 matches "1,3,7-Octatriene, 3,7-dimethyl-" aka α-Ocimene (mainlib, DMF = 925, RMF = 925), with $I$ = 1047, or "trans-β-Ocimene" (replib, DMF = 913, RMF = 914), with $I$ = 1049; then, Analyte 18 matches "β-Ocimene" as above or "1,3,6-Octatriene, 3,7-dimethyl-, (Z)-" aka cis-β-Ocimene (replib, MF = 904, RMF = 905), with $I$ = 1038. Note, Analyte 13 was used in the preliminary $I^T$ calibration, so this identification change affects the $I^T$ model. Of course, it is preferable to analyze known standards for surer identification and $I^T$ calibration. The identifications are validated by the components of *Cannabis Terpene Standards #1* from Restek (Bellefonte, PA) [16], the source sample for the chromatogram, which lists "Ocimene" (CAS 13877-91-3, β-Ocimene unspecified isomer). The 1D chromatogram for the standard supplied by Restek shows two Ocimene peaks, presumably trans-β-Ocimene (here, Analyte 13) and cis-β-Ocimene (here, Analyte 18).

3. Analyte 21 is identified as "1,6,10-Dodecatrien-3-ol, 3,7,11-trimethyl-" (Nerolidol, unspecified isomer). In replib, there is a better match (DMF = 949, RMF = 954) with "Nerolidol" aka D-Nerolidol, which has a passing $I$ = 1544 (compared to 1530 for the computed $I^T$). It is notable that the NIST 17 record for D-Nerolidol lists $I$ = 1544 ± 16, which is an exceptionally large range. Restek [16] lists Nerolidol (CAS 7212-44-4, unspecified isomer) and shows two Nerolidol peaks in the 1D chromatogram, corresponding here to Analytes 21 and 19.

Referencing Table S1 in the Supplementary Materials, as expected, the RI passing rate decreases with analyte intensity. For analytes with greater than 0.1% response, the RI passing rate is 84% (with 9% failing); for analytes with 0.02% to 0.1% response, the RI passing rate is 52% (with 28% failing); for analytes with less than 0.02% response, the RI passing rate falls to only 26% (with 45% failing).

The Base Peak test is less strongly indicative of misidentification but can help detect some erroneous identifications. Of the Top 21 analytes, shown in Table 1, six identifications failed the Base Peak check. Of those, all passed the DMF and RI checks. Some failures of Base Peak check are useful to correct misidentification; other failures may not indicate incorrect identification.

1. For Analyte 3, identified as "Linalool", there are no clear alternatives with passing $I$ and the same base peak among the top hits for the NIST libraries. However, the Wiley Registry of Mass Spectral Data (7th Edition) [17] lists 25 entries for linalool, of which 17 have base peak 71, 3 have base peak 93, and 1 has base peak 69. So, this Base Peak test failure could be due to variable EI conditions and/or mass analyzer. The identity is validated by the Restek documentation;

2. For Analyte 6, one of the spectra in replib for "Isopulegol", with a passing $I$ = 1146 (compared to 1153 for the computed $I^T$), is an even better match (DMF = 935, RMF = 942) than the #1 hit in mainlib, "(1R,2R,5S)-5-Methyl-2-(prop-1-en-2-yl)cyclohexanol" aka Neoisopulegol. In mainlib, the spectrum for "Isopulegol" was the #4 hit. The identity, Isopulegol, is validated by the Restek documentation;

3. For Analyte 10, one of the spectra in replib for the #1 hit from mainlib, "Caryophyllene" aka trans-β-Caryophyllene, has a matching base peak of 41 (DMF = 921, RMF = 930, Prob = 29). Here also, the spectra for the #1 hit are variable enough to account for the failure of the Base Peak test. The identity is validated by the Restek documentation;

4. For Analyte 14, replib has a spectrum for "1,3-Cyclohexadiene, 1-methyl-4-(1-methylethyl)-" aka α-Terpinene in which the peak for 93 is nearly as large as the base peak of 121, with intensity 906 on a scale of 999. Again, this failure may be due to somewhat different fragmentation. The identity is validated by the Restek documentation;

5. For Analyte 19, the replib has a spectrum for the #1 hit "1,6,10-Dodecatrien-3-ol, 3,7,11-trimethyl-, (E)-" aka E-Nerolidol that is a better match (DMF = 939, RMF = 946, Prob = 37) and has a matching base peak of 41. Another NIST library entry, "1,6,10-Dodecatrien-3-ol, 3,7,11-trimethyl-"

aka Nerolidol (unspecified isomer), has the same *I* and base peaks of 41 or 69 in replicate spectra. As described above, Restek [16] shows two Nerolidol peaks in the 1D chromatogram, corresponding here to Analytes 21 and 19;

6. For Analyte 20, replicate spectra for "Geraniol" have base peaks of 41 or 69. The identity is validated by the Restek documentation.

Referencing Table S1 in the Supplementary Materials, as expected, the Base Peak passing rate decreases with analyte intensity. For analytes with greater than 0.1% response, the Base Peak passing rate is 78%; for analytes with 0.02% to 0.1% response, the Base Peak passing rate is 48%; for analytes with less than 0.02% response, the Base Peak passing rate is 48%.

As seen in these examples, the results of these checks can be used to prioritize interactive examination so that the analyst can determine the final identification. For such interactive examinations, replicate spectra can be useful as well as custom MS libraries with spectra acquired under consistent conditions.

The Restek specification reveals one misidentification that passed all tests. Analyte 16 was identified as o-Cymene, but the specification lists p-Cymene. The spectra and retention indices of these two isomers are nearly identical. The #1 hit for Analyte 16 for the Wiley library was p-Cymene (DMF = 947, RMF = 947); the #1 hit for the NIST 17 library was o-Cymene (DMF = 944, RMF = 958). NIST 17 has $I = 1022 \pm 2$ for o-Cymene and $I = 1025 \pm 2$ for p-Cymene. Such fine distinctions cannot be ascertained from off-the-shelf MS libraries and require standards run under the same conditions.

The table for all 144 analytes, with identification changes described above for Analytes 6, 8, 13, 16, 18, and 21, updated $I^T$ calibration model, and MS search with both mainlib and replib, is included in the Supplementary Materials as Table S1. The AMF across all 144 analytes averaged 848, with a range from 564 to 955.

## 5. Discussion and Conclusions

For the terpene-standards chromatogram from the 2019 GC×GC Data Challenge for the Tenth Multidimensional Chromatography Workshop (Liege, Belgium), the 2D drain algorithm detected 811 blob peaks in the first 40 min of data. Of these, simple filtering on $^2t_R$ reduced the number to 413. A new method for recognizing false detections (e.g., from tails, streaks, and noise) predicted 132 true detections and 281 false detections with an accuracy rate of better than 87% (determined by interactive review). After interactive review and editing, more than 144 resolved or mostly resolved analytes were confirmed. Additionally, ion-peaks detection found 20 additional analytes, 6 additional trace compounds and 14 coeluted analytes. These methods for analyte detection found a large number of analytes in a sample which presumably was intended to have only a few standard compounds.

A new method for determining MS search settings to optimize search performance found the best performance from integrating blob spectra with TIC intensity 45% of the apex TIC and intensity thresholding at 250 DN. With the NIST mainlib, this improved average AMF for all analytes to 838, compared to 796 and 809 for simple apex and blob spectra, respectively, and improved $I^T$-fit $R^2$ for all analytes to 0.95, compared to 0.78 and 0.72 for simple apex and blob spectra, respectively. The resulting MS library-search identifications were used to extract library retention indices for the most prominent peaks, which then were pruned of inconsistencies and used for preliminary $I^T$ calibration. MS library search yielded excellent matching metrics for all of the prominent peaks, but computable identification checks are useful for flagging possible misidentifications. Three example identification checks are demonstrated. For all checks, the passing rates are positively related with analyte intensity. The identifications of the Top 21 analytes were examined, using both the NIST 17 mainlib and replib, and five identifications were changed. Finally, the $I^T$ calibration was updated and MS Search was repeated with both the NIST mainlib and replib, resulting in an average AMF of 848.

**Supplementary Materials:** The following are available online at http://www.mdpi.com/2297-8739/6/3/38/s1,
Table S1: Analyte Identifications; Table S2: Detected Ion-Peak Sets; Table S3: MS Search Experiments; Processed
chromatogram files: dataChallengeCaseStudy_20190426.[gci|bin].

**Author Contributions:** Conceptualization, S.E.R., Q.T., C.C., and C.B.; methodology, S.E.R. and Q.T.; software,
S.E.R. and Q.T.; formal analysis, S.E.R.; validation, S.E.R., Q.T., C.C., and C.B.; writing—original draft preparation,
S.E.R.; writing—review and editing, S.E.R., Q.T., C.C., and C.B.; visualization, S.E.R.

**Funding:** This research received no external funding.

**Acknowledgments:** We thank James Harynuk, University of Alberta, Canada, and Flavio A. Franchina, University
of Liège, Belgium, for creating the 2019 GCxGC Data Challenge for the Tenth Multidimensional Chromatography
Workshop (Liege, Belgium) and for providing the open data for it, including the terpene-standards chromatogram
analyzed in this case study.

**Conflicts of Interest:** S. Reichenbach and Q. Tao have a financial interest in GC Image, LLC.

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
