# Peer review of "A Data-Challenge Case Study of Analyte Detection and Identification with Comprehensive Two-Dimensional Gas Chromatography with Mass Spectrometry (GC×GC-MS)"

_separations, doi:10.3390/separations6030038_

Round 1
Reviewer 1 Report
This manuscript describes a comprehensive data analysis of a GCxGC-MS challenge data set. There is a wealth of information presented and I am pleased that the authors have chosen to publish it in open access format. This is an excellent contribution to overall understanding of GCxGC-MS data analysis. I have a few simple comments and suggestions.
The figure numbering in the text and references to figures came up strangely. This may be an artifact of my PDF viewer, but please check throughout.
Analysis of a 90 min run was stopped at 40 min. Were there any interesting analytes at higher 1D retention times? One purpose of advanced data analysis is to find these trace components.
Section 4.1 contains a lot of information and several figures. Perhaps it can be broken up into some sub-headings to help the reader follow the discussion.
In Table 2 line 2, MS should read "Mass".
Reviewer 2 Report
Journal: Separations
Title: A Data-Challenge Case Study of Analyte Detection and Identification with Comprehensive Two-Dimensional Gas Chromatography with Mass Spectrometry (GCxGC-MS)
Authors: Stephen E. Reichenbach, Qingping Tao, Chiara Cordero, and Carlo Bicchi
The authors describe their attempt to tentatively identify terpene standards (impurities?) by two-dimensional gas chromatography with mass spectrometry (GCxGC-MS) in detail. The authors have utilized the possibility of submitting supplementary material, which should be praised.
The main problem is the lack of validation; in a data challenge case study the true peaks should be known or at least the tentatively identified compound should be justified. Mass spectral library search allows only putative identification, especially the isomers cannot be distinguished.
The performance indicators are out of the range for positive identifications.
Some parts of the manuscript are rather similar to an interim report “we have not stolen the taxpayers’ money”, than a scientific investigation.
The description is too detailed still the work is hardly reproducible, which is a key factor in natural sciences
Minor errors
The letter x (iks) and multiplication sign (×) are different.
Figure 1A shows that 2D GC is superfluous.
Figure 1B does not show anything
The redundant and wordy formulations should be avoided, e.g. At page 2 dawn one reads the description of the technique, filtering etc. 3. Times.
Line 83: “As seen in Error! Reference source not found.,” – the pdf file conversion has not been checked at all. Correctly: As seen in ref [number].
“bleed steak” – the colloquial term has not been explained.
The numeration of the figures is confused: After fig. 1 fig. 3 ABC follows – all in duplicates.
No “three horizontal lines” can be seen in the figure.
There is one more figure 1 in page 5.
Alone this fundamental mismatch deserves rejection.
Figure 3 in page 6: the noise oscillations in the figures does not match with the explanation below.
“Precision v. Recall”– versus? Abbreviated as vs. It should be set in Italics.
Avoid the usage of superfluous words in the title text, etc., e.g. “case study”, “in this case study”, etc.
The abbreviation for retention indices (RI)-s is incorrect. See the IUPAC Gold Book:
https://goldbook.iupac.org/html/R/R05360.html BTW. there are more definition for retention indices.
The references are not very numerous and deficient.
The linear fit for retention indices is not adequate, the piecewise linear fit is extremely improper, see e.g. Figure 12 left part.
Avoid such empty statements as “The results of these experiments are of interest for this case study…”
I wonder what Fig. 6 in page 12 does tell us. Identification by using all analytes is worse than with using top#1 match? This is trivial.
Why the authors would like to publish such results as “there is very little difference in the results” it is simply not interesting.
“ intensity Threshold” – why the “T” is capitalized?
Etc. etc.
Bar plots are to be avoided according to the recommendations of the American Chemical Society: “In general, bar graphs are a waste of space and are discouraged.” [Anal. Chem. 2007, 79, 387–391].
Considering the above shortcomings there is no wonder I cannot recommend publication in its present form. Even a resubmission is not feasible if the minor errors are corrected only. Removing the redundancy, condensation, validation, etc, are all important to gat the manuscript be accepted.
June 07 / 2019 referee:
Reviewer 3 Report
General comments.
In the paper “A Data-Challenge Case Study of Analyte Detection 3 and Identification with Comprehensive Two4 Dimensional Gas Chromatography with Mass 5 Spectrometry (GCxGC-MS)” authors describe a case study concerning the data analysis of a chromatogram distributed for the 2019 GCxGC Data Challenge for the Tenth Multidimensional Chromatography Workshop (Liege, Belgium).
Report is good discussed but, unfortunately sometimes confused. I suggest major revisions.
Specific comments.
Abstract
Please, add a general discussion concerning some application regarding the use of mass spectrometry in environmental analyses and food characterizations. In this context add some references such as Polychlorinated biphenyls in sediments from Sicilian coastal area (Scoglitti) using automated soxhlet, GC-MS, and principal component analysis. Polycyclic Aromatic Compounds, 34(3), 237-262.
Please, in all document correct the sequent sentence with relative figure or table
“Error! Reference source not found”
Please, check and correct name of Figure. In the document are reported two Figure 3B and two Figure 3 C. On the another hand, figure 2 can be reported before than figure 3. The same concept for figure 1.
Round 2
Reviewer 2 Report
The authors have made a lot of corrections and used supplementary material extensively included new references, as well. I also read their response thoroughly, but they could not convince me.
Maybe the doc file was corrupted, but they should not “approve” the pdf file during the submission process. “We did not realize that this would happen and only discovered the problem when we read the review”, just this is the main problem, which limits the publication.
The science is not to be diligent and be paid for publication.
“The primary goal of this work was not to identify the terpene standards, but was rather to demonstrate a sequence of operations for data analysis”.
i) I do not think that to demonstrate a sequence of operations is necessary. If the primary goal was not identification, then the paper is out of scope and another publication channel is warranted.
ii) Three newly developed techniques might be suitable to another journal focused on chemometrics and data analysis.
There is no need to explain any aspect of the data challenge to the reviewer, he or she knows more about the aspects than the authors. Obtaining a list of compounds should have been done from the sponsors of the data challenge earlier.
If the authors are unsure about the meaning of “performance parameters”, how could they evaluate the performance of their methods, algorithms and procedures?
“However, the implication seems to be that there is no substantive work.” Just on the contrary the interim report is made for money, science is more, much more than to prove the diligence. The achievement should be communicated in a reproducible way. Present manuscript is far from being that.
The authors have definitely misunderstood their role in the reviewing process: “Reviewer 2 has provided many vague critiques without any substantive review of or comments about the core of the work.” The job of the reviewer is to criticize not that of the authors. If the referee could not dig to the core of the work the manuscript is definitely wrongly written and must not published.
In fact, the authors have admitted the usefulness of the critique at one point: “the noise oscillations in the figures does not match with the explanation” – Thank you to Reviewer 2 for catching this error.
I think it is not a good policy to be angry against the referee. The aims are common: to write a better, more understandable manuscript.
The authors do not want to conform with the recommendations of the American Chemical Society. However, the editor should not interfere the scientific content of a manuscript. The decisions have to be made by the authors.
The manuscript is still too long as compared to the message it carries. I cannot recommend publication in “Separations” Sorry that I cannot be more positive at the moment.
June 25 /2019 referee:
Author Response
See previous response.
Reviewer 3 Report
All suggestions were made.
Article can be accept
Ok
Author Response
Thank you.